# A high-level cloud detection method utilizing the GOSAT TANSO–FTS water vapor saturated band

Nawo Eguchi[1] and Yukio Yoshida[2]

[1]Research Institute for Applied Mechanics (RIAM), Kyushu University, Kasuga Park 6-1, Kasuga, Fukuoka, Japan
[2]Center for Global Environmental Research (CGER), National Institute for Environmental Studies (NIES), Onogawa 16-2, Tsukuba, Ibaraki, Japan

*Correspondence to:* Nawo Eguchi (nawo@riam.kyushu-u.ac.jp)

**Abstract.** A detection method for high-level clouds, such as ice clouds, is developed using the water vapor saturated channels of the solar reflected spectrum observed by the Greenhouse gases Observing SATellite (GOSAT) Thermal And Near-infrared Sensor for carbon Observation Fourier Transform Spectrometer (TANSO–FTS). The clouds detected by this method are relatively optically thin (0.01 or less) and located at high altitude. Approximately 85% of the results from this method for clouds
with cloud-top altitude above 5 km agree with the Cloud-Aerosol Lidar with Orthogonal Polarization (CALIOP) cloud classification. GOSAT has been observing since April 2009 with a 3-day repeat cycle for a pointwise geolocation pattern, providing a spectral data record that exceeds 9 years. Cloud information derived from GOSAT TANSO–FTS spectra could be powerful data for understanding the variability in cirrus cloud on temporal scales from synoptic to interannual.

**1  Introduction**

Cloud detection, especially for optically thin clouds at high altitudes, is important for enabling more accurate atmospheric trace gas retrievals from satellite remote sensing. Because cirrus cloud parameters have highly variable spatial and temporal scales, satellite observations for atmospheric trace gases require simultaneous cloud data over the same area. The cirrus cloud top altitude, which depends on tropopause height, was found to have a maximum of about 16 km in the tropics and to decrease
with increasing latitude (about 10 km in the middle latitudes) (Nazaryan et al., 2008; Sassen et al., 2008; Eguchi et al., 2007). The optical thickness of regularly observed cirrus clouds varies between 0.1 and 3.0 (Eguchi et al., 2007). Optically thin (optical thickness less than 0.1) and higher-level (above 8 km) clouds are in general difficult to detect with conventional passive sensors that measure reflected sunlight (Holz et al., 2016). However, clouds with optical thickness less than 0.05 could be detected by the MODerate Resolution Imaging Spectroradiometer (MODIS) sensor (Dessler and Yang, 2003).

Some studies have used the water vapor saturated band to detect high-level clouds, such as cirrus clouds (Gao et al., 1993, 1998, 2002, 2004). The basic principle of the water vapor saturated band method is that the upward reflectance is less over lower-level clouds and cloud-free conditions because of the strong water vapor absorption in the band. On the other hand,

large upward reflectance is observed when there are clouds in the upper troposphere because there is less water vapor above the high-level clouds (Gao et al., 1993). Therefore, the reflectance of the water vapor saturated band is used to detect upper tropospheric clouds. Gao et al. (2002) used the 1.38 $\mu$m band of the MODIS to derive the reflectance of cirrus clouds globally.

The Greenhouse gases Observing SATellite (GOSAT) measures greenhouse gases over the globe using the Thermal And Near-infrared Sensor for carbon Observation (TANSO)-Fourier Transform Spectrometer (FTS). The TANSO–Cloud and Aerosol Imager (TANSO-CAI) onboard the GOSAT is used to identify cloud and aerosol information (cloud/aerosol existence and properties, such as optical thickness and effective radius) within the TANSO–FTS instantaneous field–of–view (IFOV). The TANSO-CAI detects relatively optically thick clouds, as effectively as the MODIS on board the Terra and Aqua satellites (Ishida et al., 2011), but it has difficulty with optically thinner cirrus clouds (less than 0.1 optical thickness) because it does not have a strong water vapor absorption band and/or thermal infrared band that is effective for cirrus detection.

On the other hand, TANSO–FTS has both a strong water vapor absorption band (Band 3 of TANSO–FTS) and a thermal infrared band (Band 4). Since narrow-band spectrometry has been widely used for cloud detection, the cloud detection methods could be modified or updated for the high-resolution spectrum observed by TANSO–FTS. Someya et al. (2016) have already refined the $CO_2$ slicing method that utilizes TANSO–FTS thermal infrared spectral data. The water vapor saturated method used in the current GOSAT product is a single threshold test comparing a mean signal of the predefined 11 most $H_2O$-absorptive channels in the water vapor saturated band (5150-5200 cm$^{-1}$) with the noise (Yoshida et al., 2011). Guerlet et al. (2013) used similar threshold tests and showed that the water vapor saturated band of GOSAT can detect thin cirrus clouds with the minimum of 0.02 optical thickness.

The aim of the present study is to develop a different method from the current water vapor saturated method using the typical spectral shape of the water vapor absorption band (Band 3) of TANSO–FTS constructed by the cluster analysis. The method is validated by comparison against the Cloud-Aerosol Lidar and Infrared Pathfinder Satellite Observation (CALIPSO)/Cloud-Aerosol Lidar with Orthogonal Polarization (CALIOP) data. Section 2 describes the methodology of the high-level cloud detection method, Section 3 shows the results of the comparison study with CALIOP, and Section 4 summarizes the results and gives our conclusions.

## 2 Data and Methodology

This section describes the TANSO–FTS Level 1B (L1B) spectral data and CALIOP Level 2 cloud layer data, and the methodology for detecting scattering particles in the upper troposphere, such as high-level clouds.

### 2.1 Analysis Data

#### 2.1.1 GOSAT observation overview and spectral information

The polar orbiter GOSAT was launched on 23 January 2009 and has provided observations since April 2009 with a 3-day revisit cycle. It carries two instruments, TANSO–FTS and TANSO-CAI. The present study uses TANSO–FTS L1B spectral

data to detect high-level clouds for the ultimate purpose of reducing the error in retrieved trace gases, such as $XCO_2$ and $XCH_4$ (column-averaged dry air mole fractions of $CO_2$ and $CH_4$), which are also derived from the TANSO–FTS spectral data. The GOSAT and its two instruments are described by Kuze et al. (2009, 2016).

The TANSO–FTS has four bands (Bands 1–4), comprising three narrow bands (Bands 1–3) in the SWIR region (12,900–13,200 $cm^{-1}$, 5800–6400 $cm^{-1}$, and 4800–5200 $cm^{-1}$, respectively) and a wide TIR band (700–1800 $cm^{-1}$; Band 4), and all except Band 1 are at a spectral resolution of 0.27 $cm^{-1}$. Two polarization components of incident light (i.e., P and S polarization states) are measured separately for the three SWIR bands. The TANSO–FTS has two gain settings, high (H) and middle (M), for SWIR bands. The M-gain is used over high reflectance surfaces (e.g., desert), and the H-gain elsewhere.

The IFOV of TANSO–FTS is 15.8 mrad, which corresponds to a nadir circular footprint of about 10.5 km in diameter at sea level. The acquisition time of each measurement is 4 seconds. During nominal operation, three or five points along the cross-track direction are observed in sequence. The five-point mode was used when TANSO–FTS observation was started in April 2009, but as the stability of the pointing mechanism gradually degraded, the observation mode was changed to the three-point mode after August 2010. The TANSO–FTS also occasionally observes in targeted mode.

The high-level cloud detection method uses only Band 3P spectral data, because the signal-to-noise ratio (SNR) of Band 3P is larger than that of Band 3S in most cases. Version 161.160 of the GOSAT TANSO–FTS L1B was used in this study.

### 2.1.2 CALIPSO/CALIOP data

The CALIOP Level 2 cloud layer product version 4.10 was used for this comparison analysis (Winker et al., 2007, 2010). The CALIOP instrument is on board CALIPSO, a polar orbiter with 16-day revisit time. The CALIOP footprint (FOV) is approximately 100 m (130 $\mu$rad), observations are at about 333 m intervals (horizontal resolution), and the vertical resolution varies between 30 and 60 m, depending on the altitude range. The present study used the 5 km integrated cloud layer product (footprint size 100 m $\times$ 5 km), which includes the altitudes of cloud top and bottom, and optical thickness. CALIOP can detect thin clouds with 0.01 optical thickness (e.g. McGill et al., 2007). On the other hand, the cloud-base altitude cannot be detected for optically thick clouds with optical thickness greater than approximately 4.

We extracted cirrus cloud data derived from CALIOP from 1 January 2010 to 31 May 2013. Cirrus clouds were defined following Eguchi and Kodera (2010) as having cloud top altitude above 5 km over latitudes above 30° and above 8 km over latitudes below 30°.

## 2.2 Detection of scattering particles in the upper troposphere

Figure 1 shows a schematic of the detection of scattering particles at high altitudes (upper troposphere). The basic idea is from (Gao et al., 1993, 2002). The principle is as follows: solar radiance in the saturated water vapor absorption band is completely absorbed by water vapor in the lower troposphere when scattering particles are absent in the upper troposphere. In contrast, a large amount of scattered solar radiance in this band reaches the top-of-the-atmosphere when there is scattering particles in the upper troposphere. Therefore, high-level clouds can be detected by observing the level of upward radiance in the saturated water vapor absorption band. A saturated water vapor absorption band in the 5100–5300 $cm^{-1}$ wavenumber region is covered by

TANSO–FTS Band 3. The lower panels in Fig. 1 show examples of TANSO–FTS Band 3P spectra with and without high-level cloud. As stated above, strong reflected radiance at the saturated water vapor absorption band is indicative for the high-level cloud case.

Since atmospheric water vapor load is highly variable, surface reflected radiance might contaminate the signal under very dry conditions. To investigate the contribution of the surface reflected radiance, a radiative transfer simulation was conducted by using a line-by-line one-dimensional scalar radiative transfer model HSTAR (Nakajima and Tanaka, 1986). The absorption cross-section of water vapor was calculated by LBLRTM (Clough et al., 2005). Figure 2(a) shows the simulated spectra for cloud-free cases with different precipitable water vapor amounts. Reflected radiance increases with decreasing precipitable water vapor amount, and the wavenumber region over which absorption remains saturated contracts. Figure 2(b) shows the simulated spectra for the high-level cloud case for different cloud optical thicknesses. The spectral response patterns for these two cases are clearly different, so a wavenumber region that is sensitive to high-level cloud but insensitive to surface reflection can be determined.

A similar surface reflection issue was reported for the MODIS 1.38 $\mu$m (7246.38 cm$^{-1}$) water vapor saturated band, and for the Visible Infrared Imager/Radiometer Suite (VIIRS) the contribution of surface reflection is suppressed by narrowing the band-width (Hutchison et al., 2012). Since the spectral resolution of TANSO–FTS is higher than that of VIIRS, a more suitable set of channels can be selected. By taking the small spectral shift due to the gradual optical alignment change (Kuze et al., 2012) into account, three narrow windows (thick black lines in Fig. 2) are selected for high-level cloud detection for the present study. Figure 2 also indicates that the spectral shape appears to provide additional useful information for the detection of high-level cloud.

To obtain typical spectral shapes for Band 3P, $k$-means clustering (MacQueen , 1965) was applied to a sample of Band 3P spectra normalized so that the area under the curve is unity. More than 12,000 scenes obtained from 20 to 22 March 2010 under the condition of the solar zenith angle less than 70 degree and SNR larger than 5 were clustered into 12 groups. Figure 3 shows the mean spectral shape of each group. The number of group is given in descending order of the median brightness temperature in the 10 $\mu$m window calculated from TANSO–FTS Band 4; i.e., *Group* 1 has the highest brightness temperature and probably corresponds to a scene without high-level cloud, and *Group* 12 has the lowest brightness temperature and corresponds to a scene with optically thick high-level cloud. Utilizing these mean spectral shapes as supervised data, a flowchart of the high-level cloud detection method is given in Fig. 4. Each step is explained below and the terms used in the flowchart are described in Table 2.

We first checked the quality of the spectral data (Band 3P) to avoid anomalous spectral data. Since TANSO–FTS should point to the same location during the acquisition of an interferogram, data with an unstable viewing vector during the acquisition are treated as 'poor quality data'. When the interferogram is missing or saturated, or has spike noise, the corresponding spectral data are treated as 'poor quality data'. Distorted spectral data are detected by checking the average and standard deviation of the measured spectrum in the out-of-band-pass range. If these values deviate considerably from their nominal range, the data are treated as 'poor quality data'. Furthermore, we used data with solar zenith angle less than 90° to avoid night-side data.

Next, the Band 3P spectral data are associated with one of the 12 groups (Fig. 3) by supervised classification using the minimum distance method. The whole spectral range of Band 3P is used to calculate the Euclidian distance between the normalized Band 3P spectral data and the supervised data. The Band 3P data are assigned to the group that has the minimum Euclidian distance. In general, the minimum distance is on the order of $10^{-5}$ or less. When Band 3P spectral data are too noisy, the minimum distance value becomes large. Therefore, spectral data with minimum distance greater than $10^{-3}$ are also excluded as a low-quality spectrum and are not used for further analysis. These 'poor/low quality data' were tagged as 'missing' in the high-level cloud detection method. The spectral data with a 'missing' flag (0.5–2.5% of the whole data) were not used for further analysis.

The measurement scenes (each set of TANSO–FTS observation sounding data) are categorized as 'no elevated scattering particles' scenes or 'elevated scattering particles' scenes based on the following three threshold tests.

The threshold tests use three parameters: $S_{\mathrm{wv}}$, $S_{\mathrm{ALL}}$, and *Group* (see also Table 2) and shown in Fig. 4. The other parameters used to calculate $S_{\mathrm{wv}}$ and $S_{\mathrm{ALL}}$ are also listed in Table 2. $S_{\mathrm{wv}}$ is the ratio between the averaged radiance of selected channels of Band 3P (see Fig. 2) and *NOISE*, and $S_{\mathrm{ALL}}$ is the ratio between the averaged radiance of the whole of Band 3P and *NOISE*.

In the first threshold step (Test A), the data with $S_{\mathrm{ALL}}$ less than 3.0 are sorted as 'no elevated scattering particles', because the weak intensity in the water vapor saturated channels means there are definitely no clouds at upper levels.

At the second step (Test B), data with $S_{\mathrm{ALL}}$ greater than 3.0 are divided into two classes: 'no elevated scattering particles' for $S_{\mathrm{wv}}$ less than 0.5 and 'elevated scattering particles' for $S_{\mathrm{wv}}$ greater than 2.8. The values of the thresholds are derived from Fig. 5. Figure 5 shows the number of data, the score and accumulated score of $S_{\mathrm{wv}}$ for distances between TANSO–FTS and CALIOP up to 50 km. Here, the score is the matching ratio with the CALIOP cloud flag. From Fig. 5(b), a score of less than 0.1 corresponds to $S_{\mathrm{wv}}$ less than 0.5; i.e., approximately 90% of data with $S_{\mathrm{wv}}$ less than 0.5 have 'no elevated scattering particles'. On the other hand, from Fig. 5(c), an accumulated score larger than 0.8 corresponds to $S_{\mathrm{wv}}$ larger than 2.8; i.e., approximately 80% of the data with $S_{\mathrm{wv}}$ larger than 2.8 show 'elevated scattering particles'.

This method provides a cloud flag that indicates one of 'no elevated scattering particles (clear; hereafter treated as "clear" case for TANSO-FTS water vapor saturated band method)', 'elevated scattering particles (cloud; hereafter treated as "cloud" case for TANSO-FTS water vapor saturated band method)', or 'missing', and also the 12 groups of spectral features. At the third (final) step, Test C sorts the data with the $S_{wv}$ greater than 0.5 and less than 2.8 by the spectral shape *Group* from 1 to 12. As shown in the previous part, *Group* 1–5 are sorted for 'no elevated scattering particles (clear)' and *Group* 6–12 are sorted as 'elevated scattering particles (cloud)'.

## 3  Results

This section describes a comparison with CALIOP during 2010. Figures mainly show data from January, April, July and October in 2010.

## 3.1 Comparison with CALIOP

The criteria for match-up between TANSO–FTS and CALIOP data were within 100 km for the distance between each footprint center location and within five minutes for the observation time difference. The cloud-top altitude at the highest layer (topmost of layer) and optical thickness from the CALIOP cloud layer product were used for the analysis. Note that because the TANSO–FTS detects the accumulated reflectance of cloud layers, the TANSO–FTS radiance is not always sensitive to the topmost cloud layers.

Figure 6 shows maps of the observation points of TANSO–FTS that were matched with CALIOP observations within 100 km and the fraction of cloud flag, summarized monthly for January, April, July, and October 2010. The matched points were concentrated within the northern mid-latitudes, approximately between 30°N and 60°N. In October, there are only a few match-up points due to the change of the nominal TANSO–FTS operation mode from five- to three-point mode. The clear and cloud fractions by TANSO-FTS water vapor saturated band method changed from month to month in the ranges 44% to 66% and 33% to 53%, respectively. The clear fraction was larger than that of cloud except in April. The missing data ratio ranged from 1.0% to 6.2%, and mainly resulted from the instability of the pointing mechanism (not shown).

Figure 7 shows histograms of the 12 spectral groups obtained by the minimum distance method (left panels), and the CALIOP cloud-top altitude (middle panels) and optical thickness (right panels) for January, April, July, and October 2010. The top (bottom) panels show clear (cloud) flag cases from the TANSO–FTS water vapor saturated band method.

For the clear case (top-left panels), most data were classified as *Group* 1–5, but *Group* 7 also had a relatively large percentage (5% to 10%) of the whole clear case except in July. The cloud-top altitudes were mainly distributed in the lower troposphere, especially over the ocean, and there were maxima of cloud-top altitude frequency over land at 3 and 10 km. Optical thickness values were evenly distributed in the range 0.01 to 3.0 in the upper troposphere. In contrast, the optical thickness values above 3.0 were concentrated in the lower troposphere.

The numbers at the top of the left panels represent the matching ratio for TANSO–FTS data (clear or cloudy) against the CALIOP data. The two cases are given as a positive predictive value (i.e., TANSO–FTS and CALIOP clear case divided by all CALIOP (clear and cloud) data), which is M1 case in Table 1, and a negative predictive value (TANSO–FTS and CALIOP cloud case divided by all CALIOP (clear and cloud) data), which is M2 case in Table 1, respectively. The fraction shows that approximately 88% of the cloud cases determined by TANSO–FTS matched those from CALIOP. On the other hand, the matching ratio of the clear case was worse than that of the cloudy case, ranging from 47% to 51%, because the cloud located in the lower troposphere could not be detected by TANSO–FTS.

*Groups* 7–12 dominated in the cloudy case (bottom panels). The most frequent cloud-top altitude was located around 10 km, with the distribution falling off rapidly above 10 km and gradually below 10 km. In April and July, the cloud-top altitude also had a peak in the lower troposphere (around 2 km) over the ocean. The optical thickness in the upper troposphere varied around 3.0; however, thinner optical thicknesses less than 0.1 were also similarly distributed.

Figure 8 shows the accuracy of TANSO–FTS relative to CALIOP. The matching ratio is M3, as defined in Table 1; i.e., the fraction of all the data that is cloud or clear in both datasets. At a distance of 25 km, the matching ratio averaged for total cloud

and over all of 2010 (black solid line) was approximately 71%; the matching ratio over ocean was less than that over land for all distances and throughout the year, because the clouds in the lower troposphere could not be detected by the water vapor saturated method, as shown in Fig. 7. The poor matching ratio over the ocean during the boreal summer seems to be due to the total water vapor amount during summer which was approximately three times that in boreal winter (not shown). In addition, the poor matching ratio over the ocean was caused by the lower clouds which cover the northern Pacific during this season. The water vapor saturated method could not be detected in the lower clouds but the CALIOP could detect it. It is suggested that the water vapor amount may have an influence on the high–level cloud detection by the water vapor saturated method.

On the other hand, the matching ratio for cloud above 5 km cloud top altitude (dashed lines in Fig. 8) was approximately 86% at 25 km distance. The matching ratio decreased gradually with increasing distance between TANSO–FTS and CALIOP. The fraction above 5 km cloud top altitude is excellent because it removes the lower altitude clouds, especially over the ocean. The difference between cloud over land and ocean was not large and there was no clear seasonal dependency. It is suggested that the matching ratio was independent of both the seasonal variation of water vapor amount above 5 km altitude and the surface conditions.

## 3.2   Case study for comparison with CALIOP

Because TANSO–FTS observations have the advantage of a short revisit time (3-day cycle), the cloud variations associated with synoptic-scale phenomena can be captured. This subsection describes a case study for comparing the synoptic variation of cirrus cloud between TANSO–FTS and CALIOP.

A sudden stratospheric warming (SSW) event in the polar stratosphere induces upwelling in the tropical lower stratosphere. The upwelling causes adiabatic cooling of the lower stratosphere and the tropical tropopause Layer (TTL), and as a result, cirrus clouds in the TTL occur frequently during a SSW (Eguchi et al., 2007; Eguchi and Kodera, 2010; Eguchi et al., 2015). An SSW occurred around 25 January 2010, and cirrus cloud occurrence increased after that date (Kodera et al., 2015; Eguchi et al., 2015). Figure 9 shows the 2.5° box-averaged cirrus occurrence seven days before and after the key date (25 January), with increasing occurrence especially over the convective regions including South America, equatorial Africa, the Maritime Continent, and the western Pacific. The spatial distribution of cirrus fraction derived from the TANSO–FTS water vapor saturated band method was clearer than that from CALIOP due to higher temporal resolution (in other word less missing grid at the 2.5° grid size in the 7-days period), and the fraction from TANSO–FTS was larger approximately 1.75 times than that from CALIOP. The TANSO–FTS water vapor saturated band method could be useful for studying cirrus cloud features over a short period.

## 4   Summary and Conclusions

The main purpose of GOSAT is the observation of greenhouse gases, especially $XCO_2$ and $XCH_4$. The published standard product provides these trace gases only over cloud-free locations that are defined mainly by the TANSO-CAI cloud flag and a simplified cloud flag using the water vapor saturated band to avoid contamination due to clouds. Cloud information from the same instrument is useful for detecting the influence of cloud and retrieving the trace gases more accurately. However the

TANSO-CAI cloud flag cannot detect all cloud, especially thinner cirrus cloud, and the current operational cloud flag derived from the water vapor saturated band of the TANSO–FTS covers the high–level clouds.

Previous studies have used the water vapor saturated band to detect cirrus clouds in the upper troposphere (Gao et al., 1993, 2002, 2004). The current water vapor saturated method was roughly defined by the noise level in the water vapor saturated band (5150–5200 cm$^{-1}$). The GOSAT TANSO–FTS has a higher spectral resolution (0.27 cm$^{-1}$) than other satellite instruments, so a more precise detection method can be developed. The present utilized the typical spectral shape of Band 3 constructed by the cluster analysis.

We used the 12 groups (5 clear and 7 cloudy cases) of spectral shape of Band 3 derived from the cluster analysis together with spectral features, such as the SNR and the Euclidian distance of cluster analysis, to detect elevated particles such as cirrus clouds located in the upper troposphere. Each TANSO–FTS scene is classified into one of three categories: 'no elevated scattering particles', 'elevated scattering particles', and 'missing'. The 'missing' category mainly results from instability of the pointing mechanism.

GOSAT TANSO–FTS data matched with the CALIOP foot print within 100 km and 5 minutes were distributed mainly over latitudes between 30°N and 60°N (Fig. 6). A comparison with CALIOP revealed that the clear and cloud categories had almost the same fraction in boreal winter, the matching ratio of clear was larger than that of cloudy in the summer and autumn, and the ratio of cloudy was larger than that of clear in spring. Since there is more water vapor in boreal summer than in other seasons, the matching ratio, especially over the ocean, was worse. However, no clear seasonal variation was found in the matching ratio for data that had CALIOP cloud-top altitude above 5 km (Fig. 8). It is suggested that the water vapor saturated band method is not correlated with water vapor amount above 5 km in the mid-latitudes. In the tropics the water vapor amount decreases rapidly above 8 km; consequently, this method will detect cloud with cloud-top altitude above this level in the tropics.

A comparison with CALIOP cloud data gave an approximately 86% matching ratio for elevated particles detected by TANSO–FTS, as shown in Fig. 7 and Fig. 8. The method can detect elevated particles (cirrus clouds) located in the upper troposphere with very thin optical thickness, such as 0.1 or less. For the reference, the matching ratio D/(B+D) as defined in Table 1 as the function of CALIOP optical thickness shows 40–80% at the optical thickness below 5.0 (not shown). It shows that the new method by this study has the good potential for the cirrus clouds with various optical thicknesses. The present method detects the thinner cirrus clouds with the similar capability as the current method of the GOSAT product, however the middle layer clouds of around 5 km cloud top altitude can be detected better by the current method as shown in Fig. 10.

The water vapor saturated band method can capture in more detail the variations in cirrus cloud on the synoptic scale than those derived from CALIOP, which observes cloud and aerosol with a 16-day revisit orbit (Fig. 9). The cirrus cloud dataset derived from TANSO–FTS can be a useful additional dataset for studying cirrus clouds on synoptic to interannual temporal scales. The GOSAT data are available from April 2009 to the present; we have already accumulated data for a period exceeding nine years.

Finally, the thinner clouds, especially cirrus clouds, affect the retrieval of trace gases and are a major error source. This method should allow GOSAT to produce retrievals of greenhouse gases under thinner cloud conditions. At the results, the

thinner cloud effect might contaminate the retrieved greenhouse gases amount. The issue is beyond the scope of the present study, which will be done as the near future work.

Another satellite that targets greenhouse gases also has the same contamination with thinner clouds: the Orbiting Carbon Observatory-2 (OCO-2) project reported that the $XCO_2$ retrieval amount was improved by removing cirrus cloud contamination (D. Crisp, pers. comm.). Work in the near future aims to quantitatively investigate the effect of cirrus contamination on retrieved greenhouse gases and to improve the retrieved values.

*Data availability.* The TANSO–FTS data are available via the GOSAT Data Archive Service (GDAS) at https://data2.gosat.nies.go.jp/. The CALIOP dataset is available at https://eosweb.larc.nasa.gov/project/calipso/calipso_table/.

*Author contributions.* Eguchi designed the present study, analyzed the data and prepared the manuscript. Yoshida developed the cirrus cloud detecting method and provided the cirrus cloud flag.

*Competing interests.* The authors declare that they have no conflict of interest.

*Acknowledgements.* This study was performed within the framework of the GOSAT Research Announcement.

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

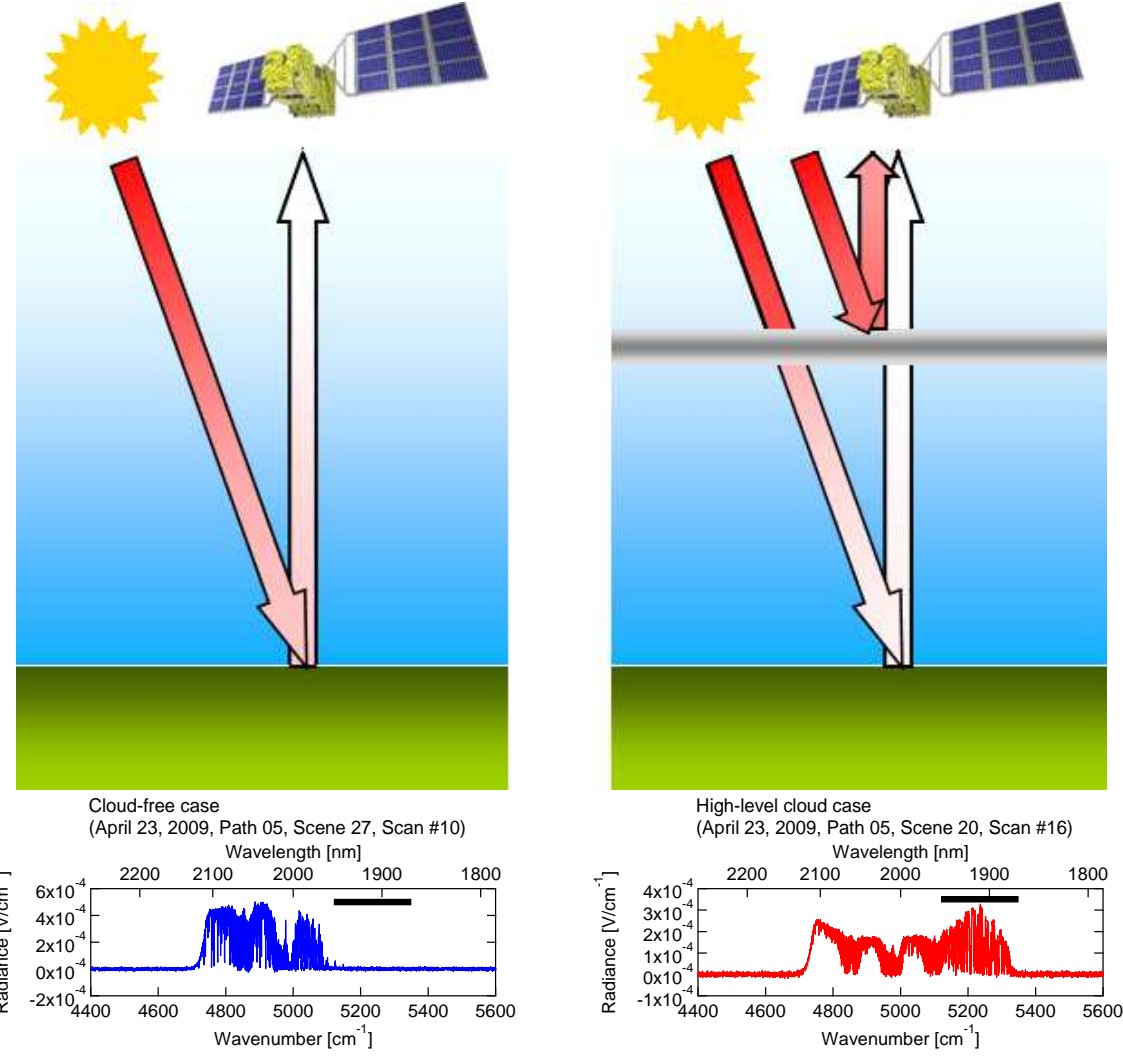

**Figure 1.** Top panels show schematics of the radiative transfer effects without (left) and with (right) scattering particles in the upper troposphere. The bottom panels show examples of the spectral shape of Band 3P without and with the scattering particles. Black thick line indicates the location of water vapor absorption band focused in this study.

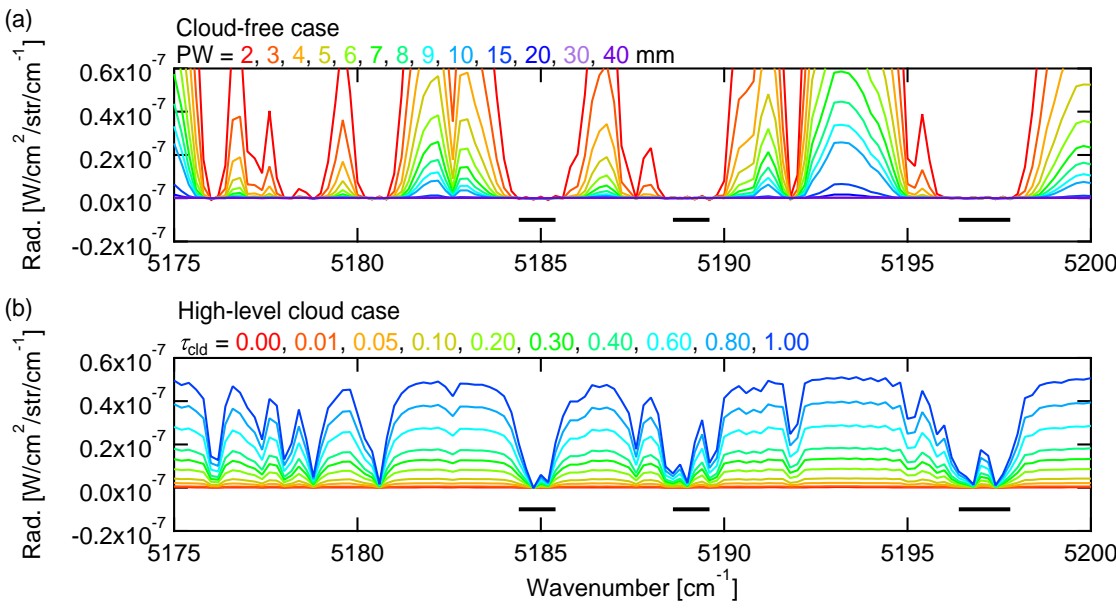

**Figure 2.** Simulated spectra for (a) cloud–free cases for different precipitable water vapor amounts and (b) high–level cloud cases for different cloud optical thicknesses. A Lambertian surface with albedo of 0.3 is assumed. Solar zenith angle and satellite zenith angle are set to $30^\circ$ and $0^\circ$ (nadir viewing), respectively. The horizontal black bars show $5184.4 - 5185.4$ cm$^{-1}$, $5188.6 - 5189.6$ cm$^{-1}$ and $5196.4 - 5197.8$ cm$^{-1}$, respectively.

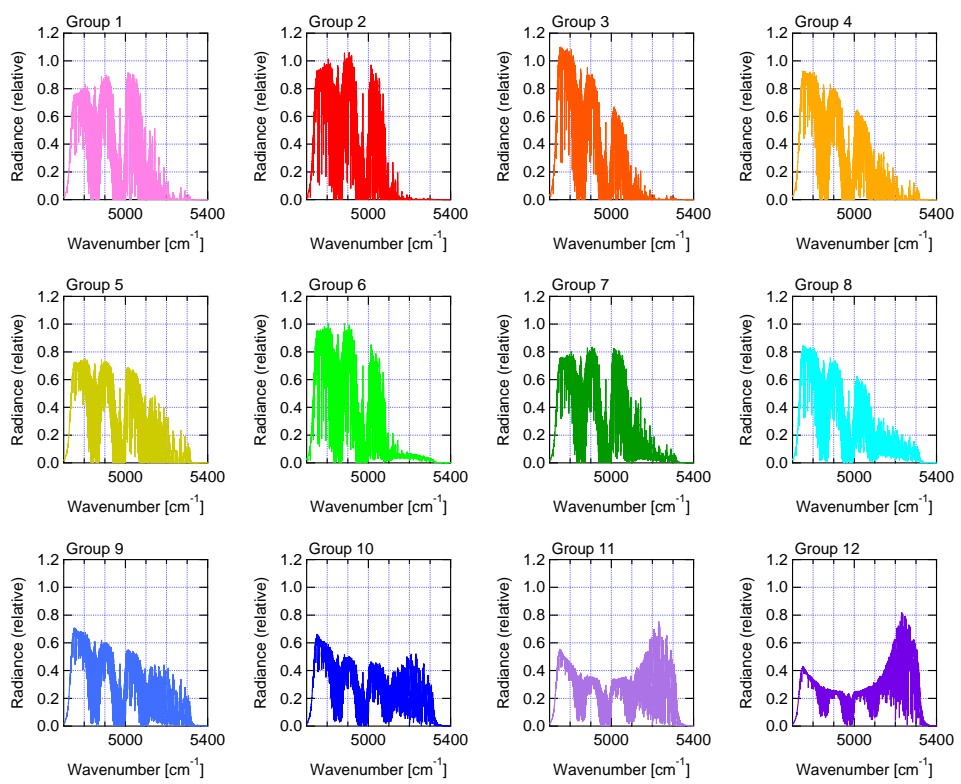

**Figure 3.** Twelve spectral groups of Band 3P spectra derived using *k*-means clustering. The *Group* from 1 to 12 were ordered from top-left to bottom-right.

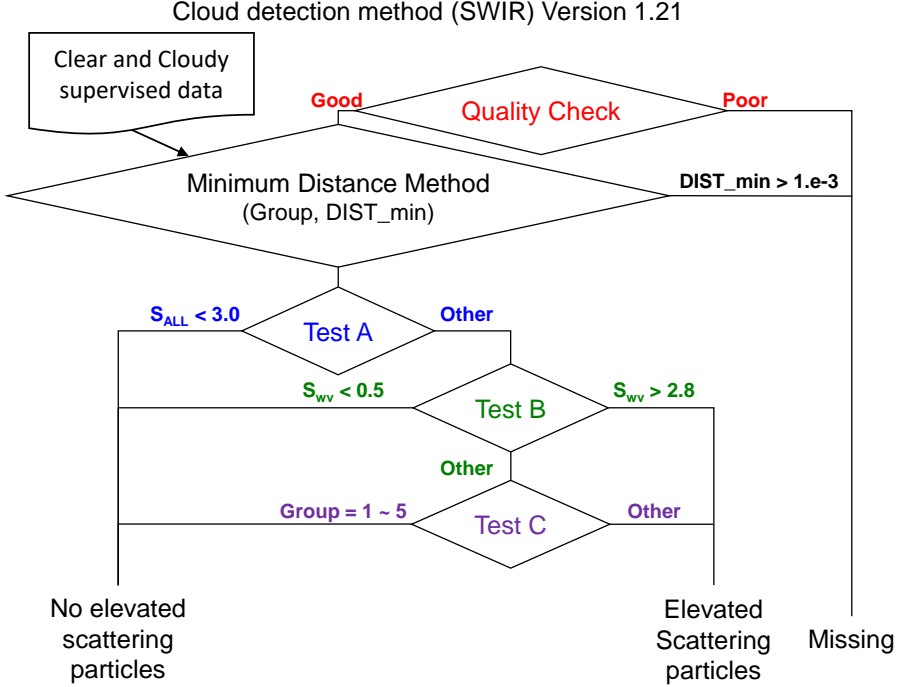

**Figure 4.** Flowchart of the method for detecting and classifying elevated scattering particles from SWIR spectra (Ver.1.21). The terms in the flowchart are described in Table 2.

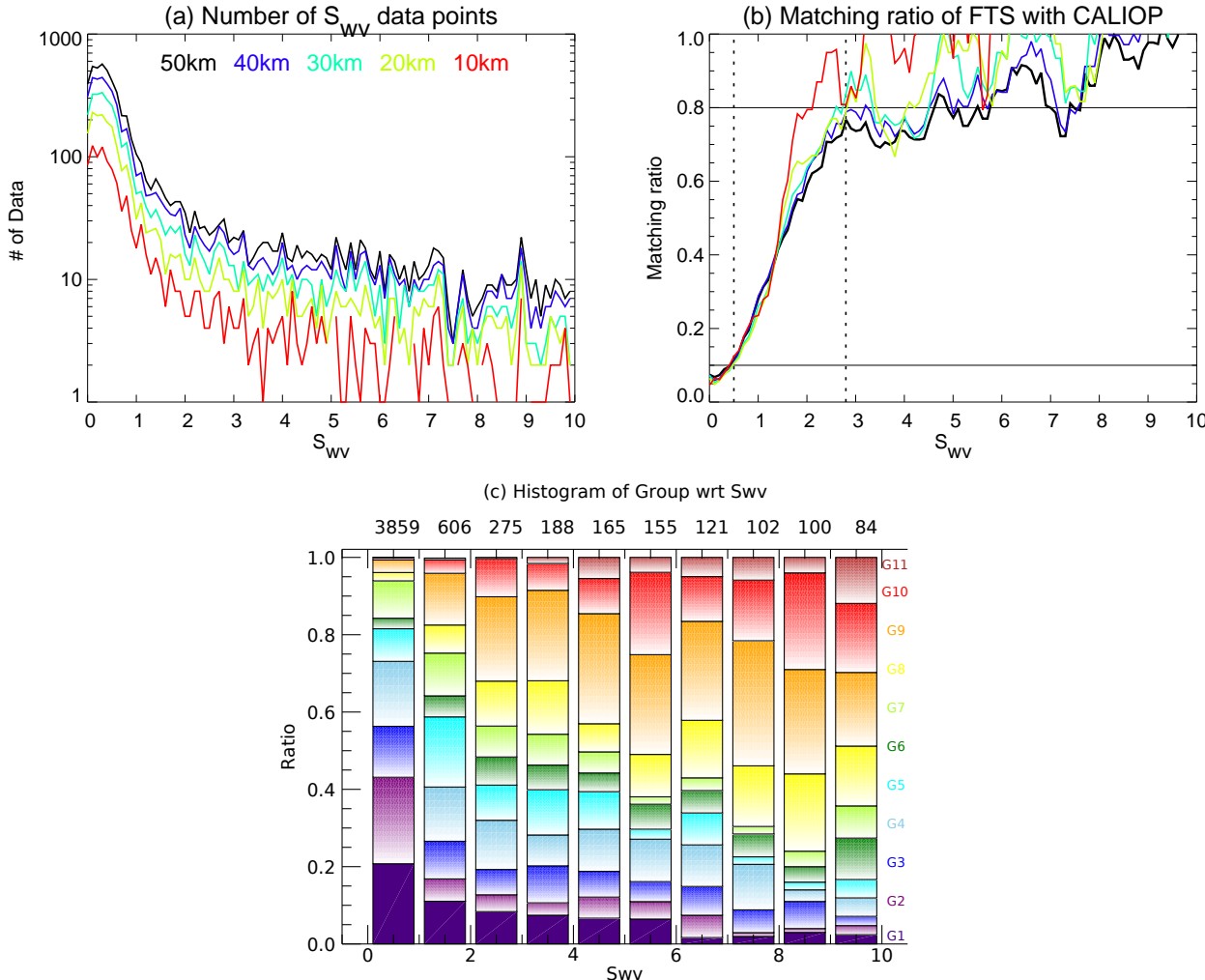

**Figure 5.** (a) Number of data points as a function of $S_{wv}$ for distance between TANSO–FTS and CALIOP within 10, 20, 30, 40, and 50 km. (b) Same as (a) but for the matching ratio between CALIOP cirrus clouds and TANSO–FTS cloud flag. (c) Histogram of group as a function of $S_{wv}$ for distance between TANSO–FTS and CALIOP within 50 km. The color bars indicate the group number from 1 to 12. *Group* 12 is absent for $S_{wv}$ less than 10.0. The numbers at the top of the graph are the total number in each bin of $S_{wv}$. The analysis period is from 1 January to 31 December 2010.

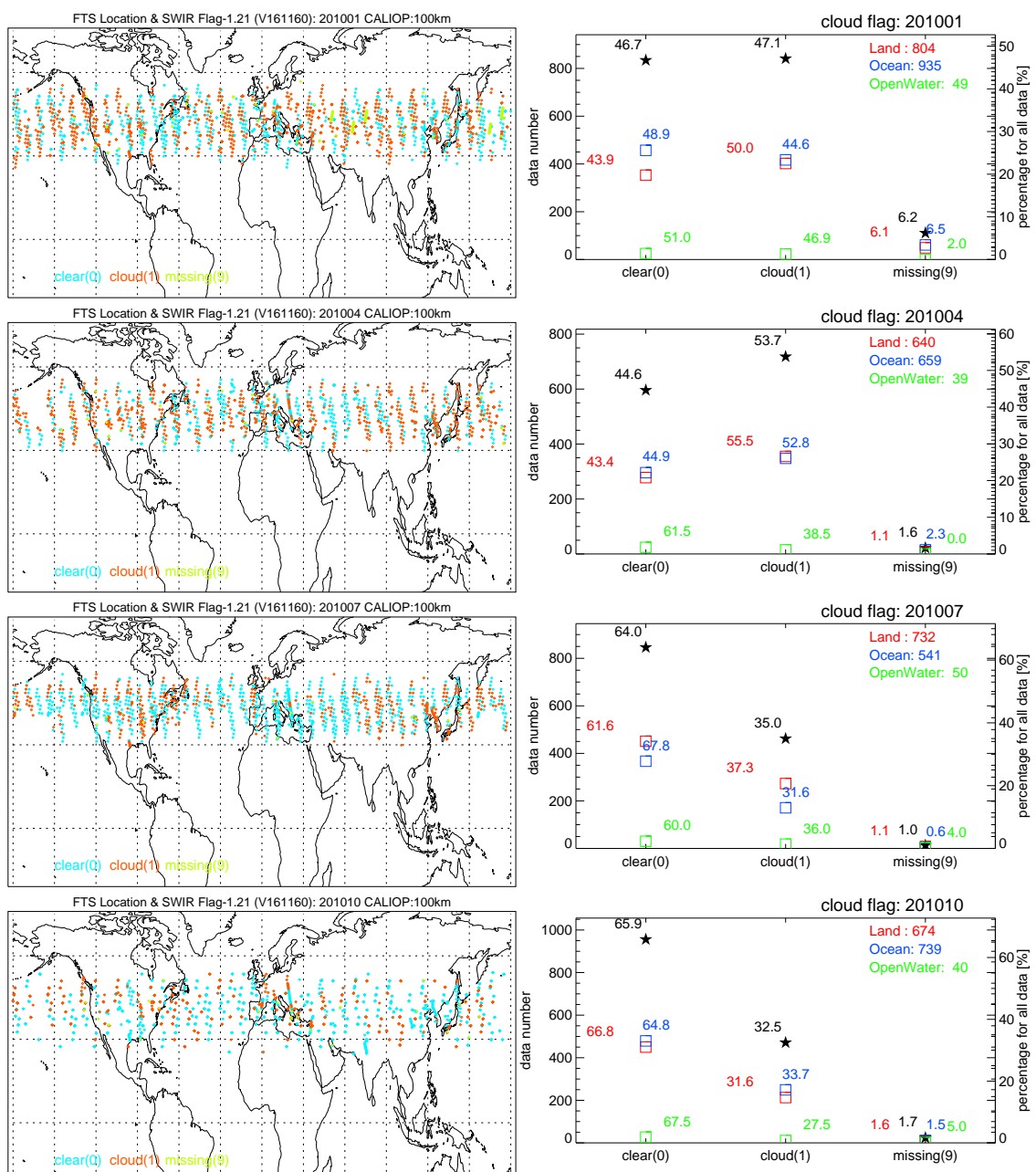

**Figure 6.** (Left) Maps of match–up observation points for TANSO–FTS and CALIOP colored according to the TANSO–FTS cloud flags: clear, cloud, and missing are shown in light blue, orange, and light green, respectively. (Right) Among the data selected for comparing with CALIOP, the fraction of each cloud flag derived from TANSO–FTS observations. The red, blue, and green colors indicate observations over land, water, and open water, respectively. The total data numbers in each case are given in the left panel. The black star symbols indicate the data number and the percentage with respect to the total dataset is also shown. The open squares in red, blue, and green show values over land, ocean, and open water, respectively, and the values to the left or right of the square are the percentage of each cloud flag with respect to the total number of data. Results are shown (from top to bottom) for January, April, July, and October 2010.

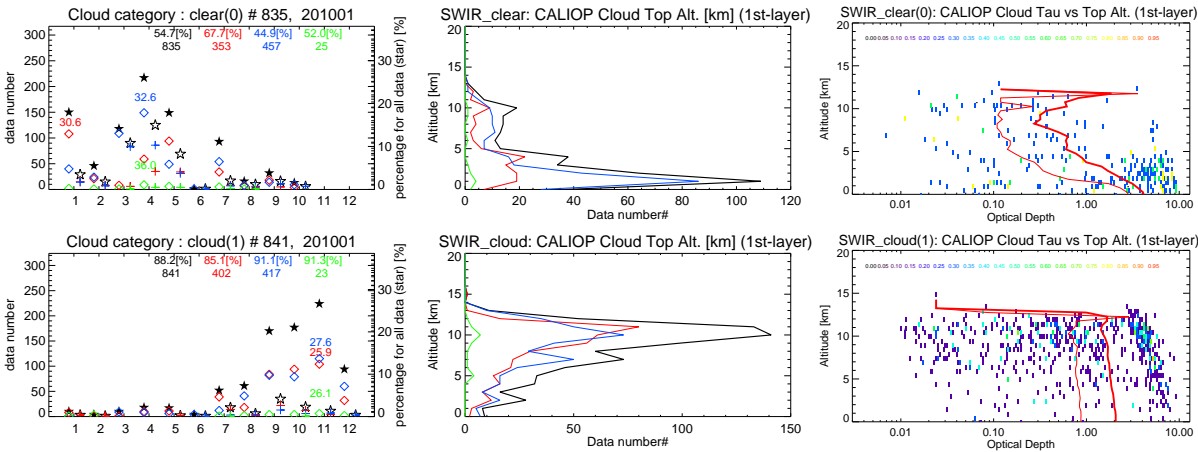

**Figure 7.** a: Top (bottom) panels show the results for clear (cloud) flags from TANSO–FTS derived by the water vapor saturated method in January 2010. (Left) Histogram of the 12 spectral groups. The closed star indicates the whole dataset, and open diamonds indicate values over land (red), ocean (blue), and open water (green). The open star and cross (+) show the histogram for data that did not match CALIOP cloud flags. The values at the top of panels show the matching ratio with CALIOP. The values near the red and blue diamonds show the highest percentage within land and ocean, respectively. (Middle) Distribution of cloud top altitude from CALIOP. The black, red, blue, and green lines show the whole dataset, land, ocean, and open water, respectively. The distance between TANSO–FTS and CALIOP is 100 km. (Right) Probability density distribution of the log–scale optical thickness from CALIOP as a function of altitude. The red thick (thin) line shows the average (median) of optical thickness at each altitude. The number at the top of panels indicates the percentage of probability density.

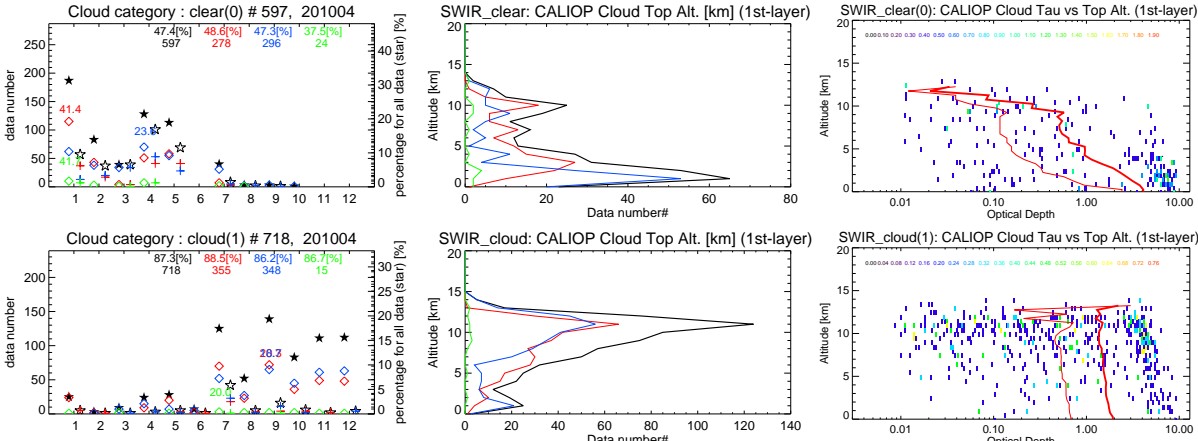

**Figure 7b.** Same as Fig. 7a but for April 2010.

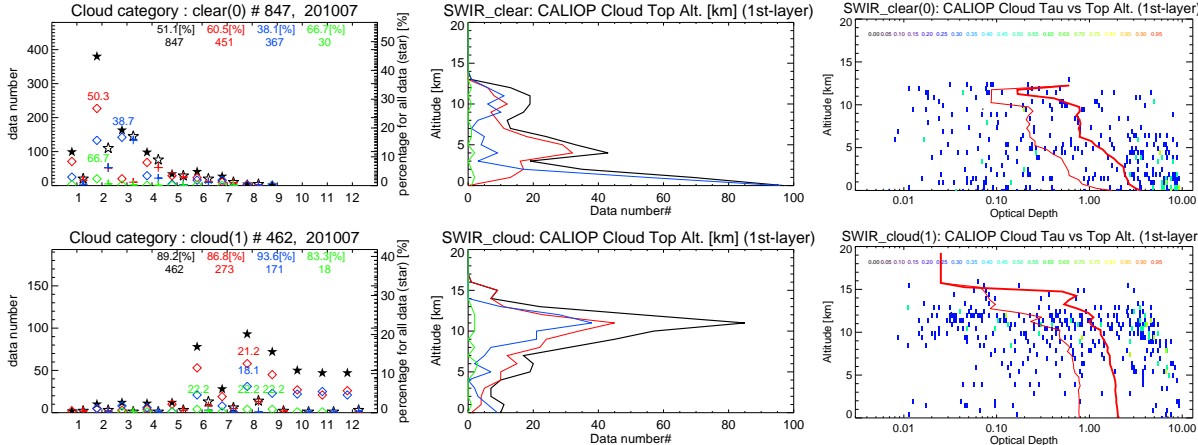

**Figure 7c.** Same as Fig. 7a but for July 2010.

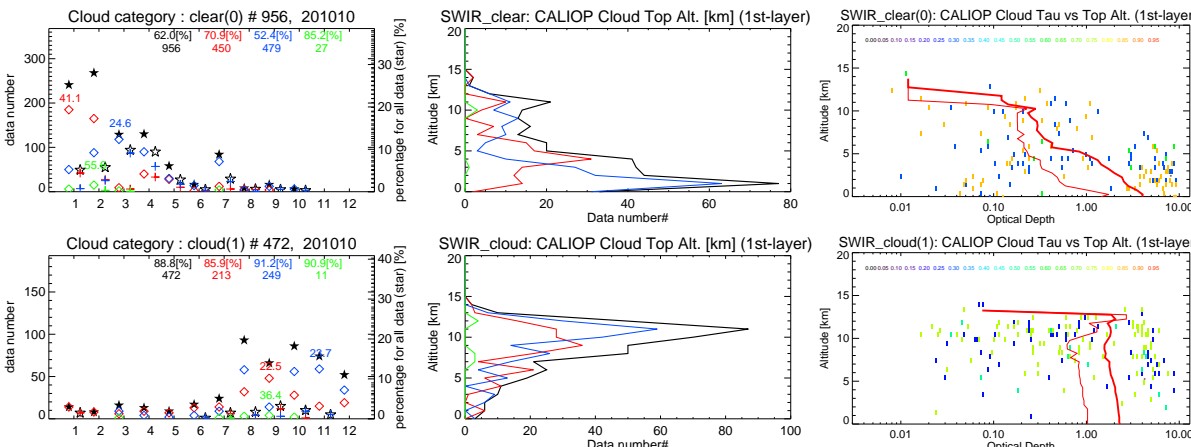

**Figure 7d.** Same as Fig. 7a but for October 2010.

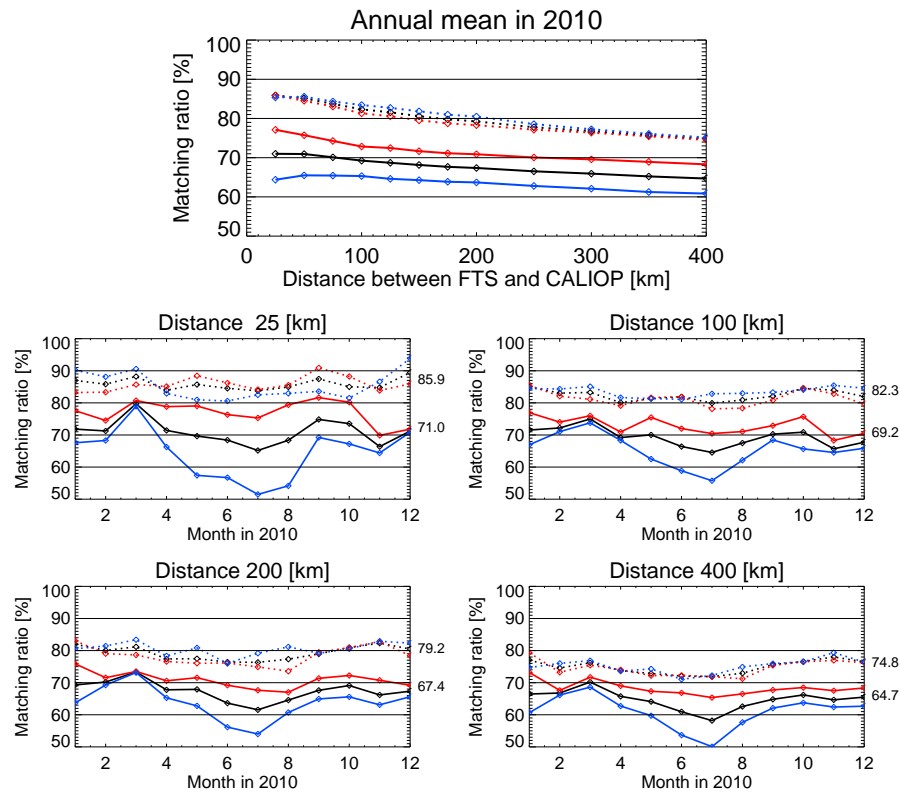

**Figure 8.** (Top panel) Annual mean matching ratio in 2010 as a function of distance. The solid lines are the whole cloud data, and the dashed lines are for cloud that has cloud top altitude above 5 km. The black, red, and blue lines indicate all data (over land, ocean, and open water), data over land, and data over ocean, respectively. (Lower four panels) Monthly matching ratio for distances between TANSO–FTS and CALIOP of 25, 100, 200, and 400 km. The values on the right–hand side of each panel show the annual means.

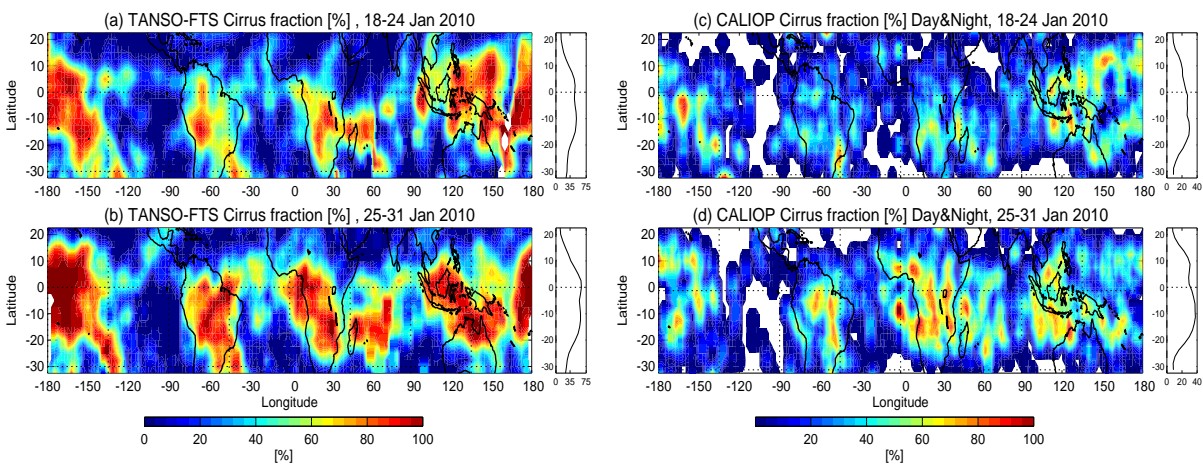

**Figure 9.** Maps of cirrus cloud occurrence frequency from the TANSO–FTS water vapor saturated method and CALIOP plotted for 2.5°
boxes. (a, c) January 18 and 24, (b, d) January 25 and 31. Both datasets were smoothed using 3–grid box smoothing. Panels to the right of
the maps show the zonal mean fraction.

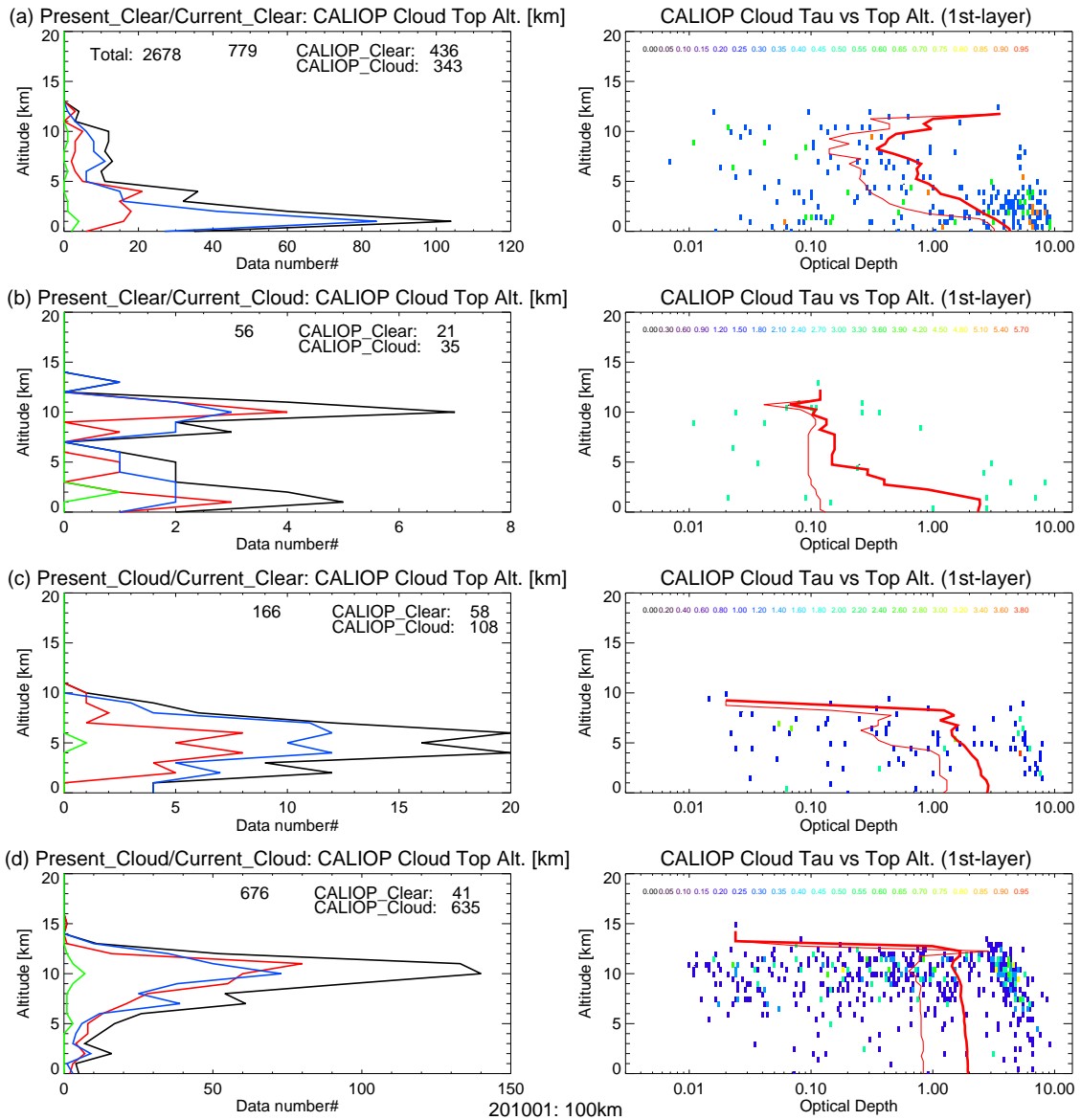

**Figure 10.** Same as the middle and right panels of Fig. 7 but for the clear and cloud cases defined by the present and current water vapor saturated methods. The distance between TANSO–FTS and CALIOP is 100 km and the period is January 2010.

**Table 1.** Definition of the matching ratio between the TANSO–FTS cloud flag and CALIOP.

|  |  | CALIOP | |
| --- | --- | --- | --- |
|  |  | clear | cloud |
| TANSO–FTS | clear | A | B |
|  | cloud | C | D |

M1 = A/(A+B)*100    M2 = D/(C+D)*100

M3 = (A+D)/(A+B+C+D)*100

**Table 2.** Terms in the flowchart shown in Fig. 4.

| Terms | Description |
| --- | --- |
| $S_{\text{wv}} = \frac{AVSPC_{\text{wv}}}{NOISE}$ | Fraction of $AVSPC_{\text{wv}}$ wrt noise level |
| $S_{\text{ALL}} = \frac{AVSPC_{\text{total}}}{NOISE}$ | Fraction of $AVSPC_{\text{total}}$ wrt noise level |
| $AVSPC_{\text{wv}} = \text{MEAN}(SPC_{\text{P}}) \begin{cases} 5184.4 \leq wn \leq 5185.4 \\ 5188.6 \leq wn \leq 5189.6 \\ 5196.4 \leq wn \leq 5197.8 \end{cases}$ | Selected channels of Band 3P. *wn* means wavenumber. |
| $AVSPC_{\text{total}} = \text{MEAN}(SPC_{\text{P}}) \begin{cases} 4400 \leq wn \leq 5700 \end{cases}$ | Averaged signal for the whole spectral region of Band 3P |
| $NOISE = 0.5 \cdot (NOISE_L + NOISE_H)$ | Averaged noises outside Band3P |
| $NOISE_L = STD(SPC_P) \begin{cases} 4450 \leq wn \leq 4600 \end{cases}$ | Standard deviation of lower (left) wavenumber outside Band3P |
| $NOISE_H = STD(SPC_p) \begin{cases} 5450 \leq wn \leq 5650 \end{cases}$ | Standard deviation of higher (right) wavenumber outside Band3P |