# Peer review of "A high-level cloud detection method utilizing the GOSAT TANSO-FTS water vapor saturated band"

_Atmospheric Measurement Techniques, 2018_

## Referee Comment (RC1) · Anonymous Referee #1 · 9 Jul 2018

The paper by Eguchi and Yoshida refines and evaluates a detection algorithm for thin cirrus clouds using saturated water vapor bands in the shortwave infrared range measured by the Greenhouse Gases Observing Satellite (GOSAT).

Variants of the method have been studied previously for other satellites (and other spectral ranges) and, variants have been used for several years within the GOSAT standard retrievals [Yoshida et al., 2011, their section 3.1] and within alternate retrievals by the science community [e.g. Guerlet et al., 2013, their section 3.2]. In particular, Guerlet et al., 2013 (their figures 4, 5, 6) did cover quite some discussion of the method. Neither Yoshida et al., 2011, nor Guerlet et al., 2013, are referenced by the manuscript.

[Figure]

While that might be negligence (Y. Yoshida is co-author here!), the question is whether the paper actually delivers some new insight. This could be argued twofold. First the algorithm is refined compared to the GOSAT standard retrievals. Second, the paper includes a comparison to CALIPSO data which has not been covered before.

So, topic-wise, the paper is suitable for Atmospheric Measurement Techniques (AMT). But scientific mass is limited. I would rate the progress compared to Yoshida et al., 2011, and Guerlet et al., 2013, as minor. Most importantly, the study (aiming at a methodological refinement) lacks a thorough quantitative comparison to the standard algorithms ie. there is no clear proof that the progress is actually positive except for some numbers in the introduction with a "not shown" attribute. There is the possibility that I just missed this key point because I could not follow the text.

The paper requires a major amount of rewording to become useful for the average AMT reader. There is a severe lack of clarity and completeness and, there is also severe language issues throughout the text; some paragraphs I simply did not understand (although I tried repeatedly).

Some detailed comments:

P2, L8+: This would be the place to refer to Yoshida et al. 2011 and to discuss the results by Guerlet et al., 2013. I do not understand what the quoted performance values against CALIOP for TANSO and CAI refer to. Is this part of the present work or is it previous work? If it is present, it appears strange to discuss it in the introduction. If it is other work, the respective references are required.

P2, L21: How can the method be applied to GOSAT-2, not even launched?

P2, L22+: This is a quite unfocused discussion of cirrus occurrence and properties which is far from complete; the number of references is small. I recommend focusing on thin cirrus and their importance because of their ubiquity in general and their impact on gas retrievals from satellites in particular.
P3, L1+: It should be made clear that the paper improves and evaluates existing methods; it does not invent the principle.

P4, L31+: If I understand correctly, the method builds on the existance of a thermal infrared channel to group the shortwave infrared radiances accoring to the thermal brightness temperature of the scene. This must be made clearer, since it implies that it would not be applicable to most of the upcoming greenhouse gas satellites (except GOSAT-2) that lack the thermal infrared. Thermal infrared brightness temperatures are representative for the layer where the remaining optical depth toward top-of-the-atmosphere is about unity. So, grouping according to thermal infrared brightness temperatures should miss thin cirrus clouds. Is this a basic flaw of the method?

P5, L2+: What does it mean: "Using these typical spectral shapes as supervised data"? Add a more detailed description of the procedure. What is "supervised data"? Why are the entire band 3P spectra used and not just the saturated micro-windows identified in Fig. 2?

P5, end of section 2: What are the fractional contributions of the tests A,B,C? Is test C actually required?

P6, L16; Fig. 6: Why do the clear cases dominate? At a footprint of 10 km of the TANSO, the expected clear-sky fraction should be much less than 10

P6-7: The whole section 3.1 is hard to follow. I do not really understand sentences such as " The poor matching ratio over the ocean, especially during the boreal summer, seems to be due to the total water vapor amount during summer was approximately three times that in boreal winter (not shown) in addition to the lower clouds could be not detected." There is several grammar mistakes and in the end, I got lost. It continues "It is suggested that the reflection by lower clouds below 5 km and the water vapor amount may have an influence on the high–level cloud detection by the water vapor saturated method." Isn't that an obvious statement given figure 2?

P6-7: The reason why this paper might be worth publishing is that the refined cirrus detection method is superior to the GOSAT standard algorithm. There appears to be an attempt to show that in appendix A but doesn't it rather show that the old algorithm has similar (or even better) performance as (than) the new one?

P7, L32: "It is clear that the TANSO–FTS water vapor saturated band method is superior for studying cirrus cloud features over a short period." A case study for 7 days without any independent validation does not support such far-reaching statements.

References

Yoshida, Y., Ota, Y., Eguchi, N., Kikuchi, N., Nobuta, K., Tran, H., Morino, I., and Yokota, T.: Retrieval algorithm for $CO_2$ and $CH_4$ column abundances from short-wavelength infrared spectral observations by the Greenhouse gases observing satellite, Atmos. Meas. Tech., 4, 717-734, https://doi.org/10.5194/amt-4-717-2011, 2011.

Guerlet, S., et al., Impact of aerosol and thin cirrus on retrieving and validating XCO2 from GOSAT shortwave infrared measurements, J. Geophys. Res. Atmos., 118, 4887–4905, doi:10.1002/jgrd.50332, 2013.

---

## Referee Comment (RC2) · Anonymous Referee #2 · 30 Jul 2018

Manuscript number: amt-2018-122 Full title: A high-level cloud detection method utilizing the GOSAT TANSO-FTS water vapor saturated band Author(s): Eguchi and Yoshida

The paper describes an improved method to detect high-level clouds using GOSAT TANSO–FTS spectral measurements. The cloud detection method is evaluated with collocated CALIOP measurements. The matching ratio of clouds (i.e., the ratio of cloud detected by both GOSAT and CALIPSO to clouds detected by GOSAT) is 85% for pixels with the collocated distance between GOSAT and CALIPSO less than 25 km. The cloud flag based on the proposed method is useful to investigate spatial distribution of

cirrus clouds. The topic in the presented paper is suitable to the journal, Atmospheric Measurement Techniques. However, the presented paper contains many grammatical mistakes and ambiguity throughout the paper. Although the authors mention the issue in thin cirrus cloud detection in the introduction, the evaluation of the cloud detection is based on all condition including low water clouds, and it is hard to know if the proposed method improves the cloud detection for thin cirrus clouds compared to the existing method. Overall, major revisions with more quantitative analysis are necessary to make the paper being worth publishing to the journal. In particular, quantitative and more detailed analysis against the following question may be needed: How much does the proposed method improve an accuracy of cloud detection compared to the existing water vapor saturation method?

Specific comments:

Page 1, line14 "optically thin (optical thickness less than 1.0) and higher-level (above 8-15 km) clouds are in general difficult to detect with conventional passive sensors that measure reflected sunlight" This statement is incorrect. Conventional passive satellite measurements can detect cloud with optical thickness of greater than 0.1–0.3 (e.g., Holz et al., 2016, Fig. 1).

Page 2, lines 11–16: There are two not show figures, which are seems to be important in terms of the motivation of this study written in lines 19–20 "Therefore the above 20 results suggest that a more precise cloud detection method to remove the higher-level clouds is required". These figures should be shown in the manuscript to enhance clarity.

Page 4, line 19: "Figure 2(a) shows the simulated spectra for cloud-free cases with different precipitable water vapor amounts." Which forward model do you use? The authors should describe at least a brief explanation about the model to reduce the ambiguity.

Page 4, line 32 to page 5, line 2, and Figure 3: What is the criterion to collect these
scenes? Are these scenes from all pixels? If so, it is hard to relate these groups with particular atmospheric conditions. For example, thin cirrus clouds and low-level marine stratocumulus clouds have similar brightness temperature (BT). Even a clear-sky with relatively low sea surface temperature and sub-visual cirrus clouds may be similar BTs. In addition, I suggest the authors to specify the mean and standard deviation of BT for each group.

Page 5, line 14: How much does the quality filter screen out low-quality data?

Page 6, lines 7-8: "The criteria for match-up between TANSO–FTS and CALIOP data were within 400 km for the distance between each footprint center location and within five minutes for the observation time difference." 400 km is too far and inappropriate for collocated-pixel comparisons. The authors use just four months of data for the comparison. Since GOSAT and CALIPSO has long measurement duration, I suggest the authors to extend the data period and to use more strict and appropriate criterion for the distance in between both satellite pixels.

Figure 7: Please specify the definition of the matching ratio for each row. In addition, if you use the CALIPSO cloud detection as a benchmark, the matching ratio M1 and M2 are odd. To show the performance of the proposed cloud detection method with TANSO-FTS, it should be M1 = A/(A+C) and M2 = D/(B+D) shouldn't be? Otherwise, we don't evaluate how much TANSO-FTS misses thin cirrus clouds that CALIPSO detect.

Page 7, lines 7–20: Does the proposed method improve cirrus cloud detection compared to the current method? What is the minimum optical thickness that the proposed method can detect clouds with an acceptable accuracy? Why do you include pixel with low stratocumulus clouds detected by CALIPSO into the analysis?

Page 8, lines 26–28: I'm not convinced with the statement. In order to state this, I suggest the authors to demonstrate at least the mating ratios (A+D)/ALL or B/(B+D) as function of cloud optical thickness.

AMTD
Reference: Holz, R. E. et al. (2016), Resolving ice cloud optical thickness biases between CALIOP and MODIS using infrared retrievals, Atmos. Chem. Phys., 16(8), 5075–5090.

---

## Author Comment (AC1) · 14 Nov 2018

Thank you very much for reviewing our manuscript. The authors understood the major points pointed out by reviewer#1. Following the comments and suggestions from two reviewers, most of them were corrected and modified, including the figures. The English was checked by a native speaker. The major revised points were the following: From the suggestion from reviewers, we did the additional analysis which fixed the program bug. The additional results showed that the water vapor saturated method from the current product of GOSAT and this study were similar, except the middle layer cloud were detected better by the new method by this study (Figure 10 in the revised

manuscript). That means both methods from the current method of GOSAT product and the new method in the present study can detect the thinner cirrus clouds of about 85% cloud frequency compared with CALIOP. Following the above new result, the introduction and summary sections were rewritten significantly. As suggested by the reviewer, some figures were modified or added. The results figures (Figs.6 and 7) were replaced for the 100 km results. Figure 4 was revised; the "Clear and Cloudy supervised data" was added to the part of "Minimum Distance Method" decision. The red line of Figure 7 was corrected; the previous red line was inconsistent with the caption and showed the summation of probability density at each altitude. Figure 10 was added to show the capability (performance) of the new method and to compare the current and new methods. We hope that these revision will be satisfied your comments.

The paper by Eguchi and Yoshida refines and evaluates a detection algorithm for thin cirrus clouds using saturated water vapor bands in the shortwave infrared range measured by the Greenhouse Gases Observing Satellite (GOSAT). Variants of the method have been studied previously for other satellites (and other spectral ranges) and, variants have been used for several years within the GOSAT standard retrievals [Yoshida et al., 2011, their section 3.1] and within alternate retrievals by the science community [e.g. Guerlet et al., 2013, their section 3.2]. In particular, Guerlet et al., 2013 (their figures 4, 5, 6) did cover quite some discussion of the method. Neither Yoshida et al., 2011, nor Guerlet et al., 2013, are referenced by the manuscript. While that might be negligence (Y. Yoshida is co-author here!), the question is whether the paper actually delivers some new insight. This could be argued twofold. First the algorithm is refined compared to the GOSAT standard retrievals. Second, the paper includes a comparison to CALIPSO data which has not been covered before. So, topic-wise, the paper is suitable for Atmospheric Measurement Techniques (AMT). But scientific mass is limited. I would rate the progress compared to Yoshida et al., 2011, and Guerlet et al., 2013, as minor. Most importantly, the study (aiming at a methodological refinement) lacks a thorough quantitative comparison to the standard algorithms ie. there is no clear proof that the progress is actually positive except for some numbers in the introduction with a

"not shown“ attribute. There is the possibility that I just missed this key point because I could not follow the text. The paper requires a major amount of rewording to become useful for the average AMT reader. There is a severe lack of clarity and completeness and, there is also severe language issues throughout the text; some paragraphs I simply did not understand (although I tried repeatedly).

Some detailed comments: P2, L8+: This would be the place to refer to Yoshida et al. 2011 and to discuss the results by Guerlet et al., 2013. I do not understand what the quoted performance values against CALIOP for TANSO and CAI refer to. Is this part of the present work or is it previous work? If it is present, it appears strange to discuss it in the introduction. If it is other work, the respective references are required. Reply: As reviewers suggested, the existing water vapor saturated method by Yoshida et al. (2011) and Guerlet et al. (2013) are described in the revised manuscript (page 2, line 15-19).

P2, L21: How can the method be applied to GOSAT-2, not even launched? Reply: The cloud detection results by this method will be used as one of the pre-screening items of the gas retrievals, although this method is not implemented in the GOSAT-2 operational processing system yet. The sentence was removed from the revised manuscript to avoid the confusion.

P2, L22+: This is a quite unfocused discussion of cirrus occurrence and properties which is far from complete; the number of references is small. I recommend focusing on thin cirrus and their importance because of their ubiquity in general and their impact on gas retrievals from satellites in particular. Reply: As reviewers suggested, the sentences were removed from the revised manuscript. The introduction was revised significantly to focus on the aim of this study.

P3, L1+: It should be made clear that the paper improves and evaluates existing methods; it does not invent the principle. Reply: We discussed the comparing analysis between the current method of the GOSAT product and the present study's method

in the summary section: the present method has the similar performance (capability) to detect the higher level clouds though comparison with CALIOP cloud data. On the other hand, the new method in this study can detect the middle layer clouds better than the current method of the GOSAT product.

P4, L31+: If I understand correctly, the method builds on the existance of a thermal infrared channel to group the shortwave infrared radiances accoring to the thermal brightness temperature of the scene. This must be made clearer, since it implies that it would not be applicable to most of the upcoming greenhouse gas satellites (except GOSAT-2) that lack the thermal infrared. Thermal infrared brightness temperatures are representative for the layer where the remaining optical depth toward top-of-the-atmosphere is about unity. So, grouping according to thermal infrared brightness temperatures should miss thin cirrus clouds. Is this a basic flaw of the method? Reply: No. We don't use the thermal infrared channel for grouping. After the grouping, we checked the median brightness temperature for each group and reordered the groups.

P5, L2+: What does it mean: "Using these typical spectral shapes as supervised data"? Add a more detailed description of the procedure. What is "supervised data"? Why are the entire band 3P spectra used and not just the saturated micro-windows identified in Fig. 2? Reply: The detail method is described in p.5 l.11-17 (p.5, l.9-16) of the previous (revised) manuscript. To avoid the confusion, several sentences were rephrased. "Mean spectral shapes" shown in Fig. 3 are used as the "supervised data". Then we classify an observed spectrum by the minimum distance method. This classification result is used in Test C. It's hard to identify from only the saturated micro-window signal, therefore, the entire band 3P spectrum is used.

P5, end of section 2: What are the fractional contributions of the tests A,B,C? Is test C actually required? Reply: The below figure shows each number divided by the Test A, B and C by the data which matched CALIOP the distance of 100 km in 2010. The results from the other distance from 25 to 400 km were an almost similar fraction. Test C is required because it treats ∼25% of input data.

Figure R1a: The data number at each test by using the data matching with CALIOP with the distance of 100 km in 2010.

P6, L16; Fig. 6: Why do the clear cases dominate? At a footprint of 10 km of the TANSO, the expected clear-sky fraction should be much less than 10 Reply: We are sorry for make you confused. Here, "clear" and "cloudy" cases mean "no elevated scattering particle" and "elevated scattering particle" cases, respectively. As discussed later, this method overlooks low-level clouds, therefore, clear (cloudy) fractions show apparently high (low) values.

P6-7: The whole section 3.1 is hard to follow. I do not really understand sentences such as " The poor matching ratio over the ocean, especially during the boreal summer, seems to be due to the total water vapor amount during summer was approximately three times that in boreal winter (not shown) in addition to the lower clouds could be not detected.“ There is several grammar mistakes and in the end, I got lost. It continues "It is suggested that the reflection by lower clouds below 5 km and the water vapor amount may have an influence on the high–level cloud detection by the water vapor saturated method.“ Isn't that an obvious statement given figure 2? Reply: Sorry for make you confusing. We rewrote the sentences clearly (p.7, l.11 - 15)

P6-7: The reason why this paper might be worth publishing is that the refined cirrus detection method is superior to the GOSAT standard algorithm. There appears to be an attempt to show that in appendix A but doesn't it rather show that the old algorithm has similar (or even better) performance as (than) the new one? Reply: The capability (performance) of the new water vapor saturated method was similar to the current method used in the present GOSAT data, although that was defined roughly. These results were added in the summary section of the revised manuscript. We believe that the results and the comparing analysis of both current method of GOSAT product and the new method by this study with CALIOP give new insight.

P7, L32: "It is clear that the TANSO–FTS water vapor saturated band method is

superior for studying cirrus cloud features over a short period." A case study for 7 days without any independent validation does not support such far-reaching statements. Reply: The sentence was rewritten as follow: "The TANSO–FTS water vapor saturated band method could be useful for studying cirrus cloud features over a short period." (p.7, l.33-34) Č

Please also note the supplement to this comment:
https://www.atmos-meas-tech-discuss.net/amt-2018-122/amt-2018-122-AC1-supplement.pdf

[Figure]

**Fig. 1.** Figure R1a: The data number at each test by using the data matching with CALIOP with the distance of 100 km in 2010.

---

## Author Comment (AC2) · 14 Nov 2018

Thank you very much for reviewing our manuscript. The authors understood the major points pointed out by reviewer#2. Following the comments and suggestions from two reviewers, most of them were corrected and modified, including the figures. The English was checked by a native speaker. The major revised points were the following: From the suggestion from reviewers, we did the additional analysis which fixed the program bug. The additional results showed that the water vapor saturated method from the current product of GOSAT and this study were similar, except the middle layer cloud were detected better by the new method by this study (Figure 10 in the revised

manuscript). That means both methods from the current method of GOSAT product and the new method in the present study can detect the thinner cirrus clouds of about 85% cloud frequency compared with CALIOP. Following the above new result, the introduction and summary sections were rewritten significantly. As suggested by the reviewer, some figures were modified or added. The results figures (Figs.6 and 7) were replaced for the 100 km results. Figure 4 was revised; the "Clear and Cloudy supervised data" was added to the part of "Minimum Distance Method" decision. The red line of Figure 7 was corrected; the previous red line was inconsistent with the caption and showed the summation of probability density at each altitude. Figure 10 was added to show the capability (performance) of the new method and to compare the current and new methods. We hope that these revision will be satisfied your comments.

Manuscript number: amt-2018-122 Full title: A high-level cloud detection method utilizing the GOSAT TANSO–FTS water vapor saturated band Author(s): Eguchi and Yoshida

The paper describes an improved method to detect high-level clouds using GOSAT TANSO–FTS spectral measurements. The cloud detection method is evaluated with collocated CALIOP measurements. The matching ratio of clouds (i.e., the ratio of cloud detected by both GOSAT and CALIPSO to clouds detected by GOSAT) is 85% for pixels with the collocated distance between GOSAT and CALIPSO less than 25 km. The cloud flag based on the proposed method is useful to investigate spatial distribution of cirrus clouds. The topic in the presented paper is suitable to the journal, Atmospheric Measurement Techniques. However, the presented paper contains many grammatical mistakes and ambiguity throughout the paper. Although the authors mention the issue in thin cirrus cloud detection in the introduction, the evaluation of the cloud detection is based on all condition including low water clouds, and it is hard to know if the proposed method improves the cloud detection for thin cirrus clouds compared to the existing method. Overall, major revisions with more quantitative analysis are necessary to make the paper being worth publishing to the journal. In particular, quantitative and

more detailed analysis against the following question may be needed: How much does the proposed method improve an accuracy of cloud detection compared to the existing water vapor saturation method?

Specific comments: Page 1, line14 "optically thin (optical thickness less than 1.0) and higher-level (above 8-15 km) clouds are in general difficult to detect with conventional passive sensors that measure reflected sunlight" This statement is incorrect. Conventional passive satellite measurements can detect cloud with optical thickness of greater than 0.1–0.3 (e.g., Holz et al., 2016, Fig. 1).

Reply: Along the reviewer comments, the sentence was rewritten (Page 1, Line 16-20).

Page 2, lines 11–16: There are two not show figures, which are seems to be important in terms of the motivation of this study written in lines 19–20 "Therefore the above results suggest that a more precise cloud detection method to remove the higher level clouds is required". These figures should be shown in the manuscript to enhance clarity.

Reply: From the additional analysis, the new method by this study and the current method from GOSAT product had the similar capability (performance) to detect the thinner cirrus clouds. Because the evidence provided new insight, these results were added in the summary section. Therefore the introduction was briefly rewritten.

Page 4, line 19: "Figure 2(a) shows the simulated spectra for cloud-free cases with different precipitable water vapor amounts." Which forward model do you use? The authors should describe at least a brief explanation about the model to reduce the ambiguity.

Reply: A line-by-line one-dimensional scalar radiative transfer model HSTAR (Nakajima and Tanaka, 1986) is used. The absorption cross-section of water vapor is calculated by LBLRTM (Clough et al., 2005). These information were added (p.4, l.12-14). T. Nakajima and M. Tanaka, Matrix formulations for the radiative transfer of solar radiation

in a plane-parallel scattering atmosphere. J. Quant. Spectr. Rad. Trans., 35, 13-21, 1986. S.A. Clough, M.W. Shephard, E.J. Mlawer, J.S. Delamere, M.J. Iacono, K. Cady-Pereira, S. Boukabara, P.D. Brown, Atmospheric radiative transfer modeling: a summary of the AER codes. J. Quant. Spectr. Rad. Trans., 91, 233-244, 2005.

Page 4, line 32 to page 5, line 2, and Figure 3: What is the criterion to collect these scenes? Are these scenes from all pixels? If so, it is hard to relate these groups with particular atmospheric conditions. For example, thin cirrus clouds and low-level marine stratocumulus clouds have similar brightness temperature (BT). Even a clear-sky with relatively low sea surface temperature and sub-visual cirrus clouds may be similar BTs. In addition, I suggest the authors to specify the mean and standard deviation of BT for each group.

Reply: The selection criteria are (i) the solar zenith angle is less than 70 deg. and (ii) SNR is larger than 5. As for the BT, it is just used for reordering the groups after classification by the k-mean clustering method. BT itself is not used in both the k-mean clustering and this cloud detection method, therefore, the mean and standard deviation of BT for each group are not shown in the revised paper.

Page 5, line 14: How much does the quality filter screen out low-quality data?

Reply: 0.5 – 2.5% data were removed as poor quality data. The result was added in the revised manuscript (p.5, l.15-16).

Page 6, lines 7-8: "The criteria for match-up between TANSO–FTS and CALIOP data were within 400 km for the distance between each footprint center location and within five minutes for the observation time difference." 400 km is too far and inappropriate for collocated-pixel comparisons. The authors use just four months of data for the comparison. Since GOSAT and CALIPSO has long measurement duration, I suggest the authors to extend the data period and to use more strict and appropriate criterion for the distance in between both satellite pixels.

Reply: As suggested by the reviewer, the results figures were replaced for the 100 km results. We analyzed every month in 2010 and also the other distances from 25 to 400 km. In the results, the features (the cloud top altitude profile by CALIOP) were almost the same in four months (Jan, Apr, Jul, and Oct) shown in the manuscript, although the matching ratio increased while decreasing the distance. Figure 8 in the revised manuscript shows the matching ratio with the function of distance.

Figure 7: Please specify the definition of the matching ratio for each row. In addition, if you use the CALIPSO cloud detection as a benchmark, the matching ratio M1 and M2 are odd. To show the performance of the proposed cloud detection method with TANSO-FTS, it should be M1 = A/(A+C) and M2 = D/(B+D) shouldn't be? Otherwise, we don't evaluate how much TANSO–FTS misses thin cirrus clouds that CALIPSO detect.

Reply: The matching ratios defined by A/(A+B) or D/(C+D) are easy to understand how much the contamination of the opposite scene (if the TANSO clear case are, we know how much the CALIOP cloud case) and the cloud top altitude observed by CALIOP. The cloud top altitudes were derived only by CALIOP.

Page 7, lines 7–20: Does the proposed method improve cirrus cloud detection compared to the current method?

Reply: The new method in this study has the same capacity to detect the thinner cirrus clouds as the old one (the current GOSAT product), except the middle layer clouds can be detected by the new method better than the old one. The new insights were written in the summary section.

What is the minimum optical thickness that the proposed method can detect clouds with an acceptable accuracy?

Reply: As the following figure show, clouds with optical thickness from 0.01 to 5.0 can be detected with the 40-80% matching ratio. That is, the new method in this study

could detect the thin clouds from 0.01 or 0.025.

Why do you include pixel with low stratocumulus clouds detected by CALIPSO into the analysis?

Reply: It is easy to understand that the water vapor saturated method (both the current and new method by this study) cannot detect the low-altitude clouds. If the clouds with the cloud top above 5 km were removed by CALIOP, we can understand the matching ratio which increase as shown in the dashed lines of Figure 8.

Page 8, lines 26–28: I'm not convinced with the statement. In order to state this, I suggest the authors to demonstrate at least the mating ratios (A+D)/ALL or B/(B+D) as function of cloud optical thickness.

Reply: The following figures show that the matching ratios (D/((B+D)) are from 40 to 80% at the optical thickness less than 5.0. The results added to the revised manuscript (p.8, l.28-29).

Figure R2a: The matching ratio (D/(B+D) in Table 2 of the manuscript) between TANSO-FTS and CALIOP with the function of CALIOP optical thickness at each month in 2010. The thin line shows the data number divided by 2. The distance between CALIOP and TANSO-FTS is 100 km.

Please also note the supplement to this comment:
https://www.atmos-meas-tech-discuss.net/amt-2018-122/amt-2018-122-AC2-supplement.pdf

———————————————

[Figure]

**Fig. 1.** Figure R2a: The matching ratio (D/(B+D) in Table 2 of the manuscript) between TANSO-FTS and CALIOP with the function of CALIOP optical thickness at each month in 2010.

**Supplement:**

[revised manuscript text omitted]

---

## Referee Report (RR1)

**Review: "A high-level cloud detection method utilizing the GOSAT TANSO–FTS water vapor saturated band"**

Nawo Eguchi and Yukio Yoshida

**General vote**

Acceptable after minor revision

**Referees synopsis**

This manuscript presents a method to retrieve high-level clouds from measurements of the TANSO-FTS instrument onboard GOSAT. The presented approach is able to detect optical thin clouds in high altitudes and shows good agreement with CALIOP cloud classification data.

The study describes the retrieval method in detail and despite some weaknesses in the English language and syntax, the reader is able to understand the approach. It might be useful to have a high-quality data record for high-level clouds, but I think the evaluation approach against CALIOP has some deficiencies, which should be improved before publication

**_Major review points_**

1. The criterion for co-locations of TANSO-FTS and CALIOP is a distance within 100 km. I understand that is difficult to get enough co-locations for just one year if you use a smaller distance, but for a distance of 100 km (or even 400 km as in Fig. 8) a match could be pure random. How do you explain a matching ration of nearly 70 % for a distance of 400 km (Fig. 8)? In my point of view a meaningful evaluation is done for co-locations within the distance of half a grid point and a time of five minutes. What is the reason to use this distance? The agreement with CALIOP looks quite good for distances of 25 km. Why don't you use just a longer period for comparison with a much smaller distance for co-locations?
2. The result from the case study comparison with CALIOP is that TANSO-FTS has a higher revisit time. Could you please quantify things like "was clearer" or "was larger" in more detail?
3. In parts of the manuscript it is mentioned that there is already a retrieval for high clouds from GOSAT. It would be nice to have a comparison against comparable products.
   At the moment it is hard for the reader to classify the quality of this data record compared to existing products, such as the MODIS cloud product, CLARA-A2 or HIRS cloud data from University Wisconsin.

**Minor review points**

4. The English has to be improved by a native speaker. Especially the usage of the plural.
5. Will the data be freely available? If yes, in which format, temporal and spatial resolution? What are the future plans for this data record?